# Prenatal care coverage and correlates of HIV testing in sub-Saharan Africa: Insight from demographic and health surveys of 16 countries

Oluwafemi Emmanuel Awopegba[1], Amarachi Kalu[2], Bright Opoku Ahinkorah[3], Abdul-Aziz Seidu[4,5], Anthony Idowu Ajayi[6]*

1 Economics and Business Policy Department, Nigerian Institute of Social and Economic Research, Ibadan, Nigeria, 2 Department of Sociology, Faculty of the Social Sciences, University of Ibadan, Ibadan, Nigeria, 3 The Australian Centre for Public and Population Health Research (ACPPHR), Faculty of Health, University of Technology, Sydney, Australia, 4 Department of Population and Health, University of Cape Coast, Cape Coast, Ghana, 5 College of Public Health, Medical and Veterinary Sciences, James Cook University, Townsville, Queensland, Australia, 6 Population Dynamics and Reproductive Health Unit, African Population and Health Research Centre, APHRC Campus, Nairobi, Kenya

* ajayianthony@gmail.com

**Data Availability Statement:** The 2018 DHS data for all countries underlying the results presented in the study are freely available from DHS at https://dhsprogram.com/data/available-datasets.cfm.

## Abstract

### Background

Prenatal screening of pregnant women for HIV is central to eliminating mother-to-child-transmission (MTCT) of HIV. While some countries in sub-Saharan Africa (SSA) have scaled up their prevention of MTCT programmes, ensuring a near-universal prenatal care HIV testing, and recording a significant reduction in new infection among children, several others have poor outcomes due to inadequate testing. We conducted a multi-country analysis of demographic and health surveys (DHS) to assess the coverage of HIV testing during pregnancy and also examine the factors associated with uptake.

### Methods

We analysed data of 64,933 women from 16 SSA countries with recent DHS datasets (2015–2018) using Stata version 16. Adjusted and unadjusted logistic regression models were used to examine correlates of prenatal care uptake of HIV testing. Statistical significance was set at p<0.05.

### Results

Progress in scaling up of prenatal care HIV testing was uneven across SSA, with only 6.1% of pregnant women tested in Chad compared to 98.1% in Rwanda. While inequality in access to HIV testing among pregnant women is pervasive in most SSA countries and particularly in West and Central Africa sub-regions, a few countries, including Rwanda, South Africa, Zimbabwe, Malawi and Zambia have managed to eliminate wealth and rural-urban inequalities in access to prenatal care HIV testing.

**Funding:** The authors received no specific funding for this work.

**Competing interests:** The authors have declared that no competing interests exist

## Conclusion

Our findings highlight the between countries and sub-regional disparities in prenatal care uptake of HIV testing in SSA. Even though no country has universal coverage of prenatal care HIV testing, East and Southern African regions have made remarkable progress towards ensuring no pregnant woman is left untested. However, the West and Central Africa regions had low coverage of prenatal care testing, with the rich and well educated having better access to testing, while the poor rarely tested. Addressing the inequitable access and coverage of HIV testing among pregnant women is vital in these sub-regions.

## Introduction

With the introduction of antiretroviral treatments (ART), there has been a 40% decline in new HIV infections from 2.9 million in 1997 to 1.7 million in 2019 [1]. Also, new infections have declined by 52% among children from 310 000 in 2010 to 150 000 in 2019 [1]. However, with over 150,000 new infections among children in 2018, mother to child transmission of HIV (MTCT) remains a major public health concern [1]. What is more, 61% of global new infections among children in 2019 occurred among children in sub-Saharan Africa (SSA), with western and central African countries having the highest record of children infected [1]. Success in eliminating MTCT is vastly uneven in SSA. While Southern and Eastern African countries such as Comoros, Malawi, Rwanda, South Africa, Burundi, and Uganda have seen a remarkable decline in MTCT transmission of HIV, many countries in West and Central Africa like Angola, Equatorial Guinea, the Gambia, Mali, and Niger have experienced an increase in transmission rate [2]. The intensification of the existing PMTCT program and scaling up of a nationwide program to accelerate MTCT elimination in Malawi, Zimbabwe, and South Africa have resulted in MTCT at 6-12weeks postpartum rate comparable to the 'global north' [3–7].

HIV testing before and during pregnancy is critical to initiating ART and eliminating MTCT of HIV [8, 9]. The risk of MTCT of HIV ranges from 15–45%, without ART but could be reduced to < 5% with prenatal care testing and initiation of ART [10]. Global north countries with a universal prenatal care HIV screening among pregnant women have nearly eliminated MTCT of HIV [11–14]. However, SSA countries, particularly West and Central African countries, still have a low prevalence of prenatal care HIV testing among pregnant women. The high rate of MTCT of HIV in SSA reflects gaps in the PMTCT programmes in SSA, particularly in screening and diagnoses of pregnant women. It is, however, worth noting that the health system's response and dynamics of PMTCT implementation in SSA are markedly different from country to country. For instance, in Rwanda and Uganda, it is mandatory for every pregnant mother who attends antenatal care (ANC) to test for HIV, rather than a voluntary choice of the mother [15, 16]. In South Africa and Botswana and like most SSA countries, the opt-out policy was implemented and pregnant women are educated about HIV. However, despite the acceptability and implementation of the opt out strategy [17], there is evidence a large proportion of women who received ANC are not tested for HIV in some SSA countries [2, 18].

Attendance of ANC is generally high in East and Southern African countries relative to West and Central Africa countries, thus explaining the contrasting successes recorded in PMTCT across SSA [19]. Even though HIV testing is generally available in all West African countries, the challenge remains to ensure women access prenatal care and also ensure all women who do are tested as part of prenatal care. In Nigeria, for example, about 30% of

women who received prenatal care were not tested for HIV [2]. As such, significant gaps remain in terms of universal testing of women who present for antenatal care. Nevertheless, the larger percentage of unreached women remains those who never received prenatal care.

Some of the reasons why pregnant women do not test for HIV in SSA include inaccessibility of healthcare facility [20], perceived lack of confidentiality, stigma, and discrimination [21, 22], cost, illiteracy, and inability to secure husband's permission, attitude, and skills of health workers and inadequate resources [23, 24]. Health behavioural theories such as the Health belief model (HBM) [25] and Capability, Opportunity, and Motivation Model of Behaviour (COM-B) model [26] have also highlighted barriers and enablers of uptake of health behaviours and behavioural change among individuals, including deciding to test for HIV during pregnancy. Most of these theories and models explain or predict how internal and external factors influence an individual's health behaviour. The Health Belief Model (HBM) could provide an understanding of the relationship between individual factors such as knowledge of MTCT and prenatal care uptake of HIV testing. Based on the proposition of HBM, mothers who have good knowledge of MTCT can assess their susceptibility correctly, understand the severity of MTCT of HIV, and the benefit of testing [27–29]. Lack of knowledge of MTCT could constitute a significant barrier to the uptake of HIV testing, as mothers may not see the need for testing or inaccurately assess their risk. According to the COM-B model, behaviour is a product of three necessary conditions; capability, opportunity, and motivation. Capability can be psychological (knowledge) or physical (skills), opportunity can be social (societal influences) or physical (environmental resources) while motivation can be automatic (emotion) or reflective (beliefs, intentions). COM-B model provides a comprehensive analysis of all factors influencing prenatal HIV testing [26].

Given the lack of recent data on prenatal testing in SSA and drawing from the COM-B model and HBM theoretical propositions, we examined factors associated with uptake of HIV testing during pregnancy in 16 sub-Saharan African countries. These countries are representative of all the sub-regions of SSA. Our study aligns with the global effort to achieve the UNAIDS first 95% by 2030. The findings of the study will be useful for scaling up HIV testing in SSA. It will facilitate cross-country comparison and could be used to support advocacy. In the present study, we first estimated the prevalence of HIV testing during pregnancy in SSA and the 16 countries included in our analysis. We then assessed the effects of knowledge of MTCT, relevant individual and community level factors on HIV testing in these countries.

## Methods

This cross-sectional study used recent Demographic and Health Surveys (DHS) data (2015–2018) to examine MTCT knowledge and uptake of HIV testing during pregnancy in SSA. DHS is a nationally representative survey collected every five years across low- and middle-income countries. We focused on 16 countries with recent DHS to assess the landscape of HIV testing during pregnancy in SSA. Countries were included in the study if they had complete information on HIV testing during pregnancy, MTCT knowledge, and have recent DHS (2015–2018) (see Fig 1). The countries included were Benin, Guinea, Mali and Senegal from West Africa, Angola, Cameroon and Chad from Central Africa, Burundi, Ethiopia, Rwanda and Uganda from East Africa and Malawi, Mozambique, South Africa, Zambia and Zimbabwe from Southern Africa.

The study population was women aged 15 to 49 who gave birth within two years preceding the surveys. Weightings were applied to obtain unbiased estimates, according to the DHS guidelines. Not using weights can bias results toward the oversampled sub-populations. The study sample was, thus, a weighted sample of 65,107 women (Table 1). Details on the sampling methodology and data collection used by the DHS are published elsewhere [30].

Identification
- sub-Saharan African countries with DHS datasets from 2015 to 2020 (n=18)

Screening
- Two countries (Tanzania and Nigeria) with missing data on HIV testing during pregnancy were removed from the analysis.

Inclusion
- Countries included in the analysis: Central Africa (n=3); West Africa (n=4); East Africa (n=4); and Southern Africa (n=5).

**Fig 1. Overview of country selection.**

Table 1. Sampling distribution and countries.

| Countries and sub-region | Data Collection Year | Unweighted observations remaining | Weighted observations remaining |
|---|---|---|---|
| Central Africa | | | |
| Angola | 2016 | 5,631 | 5,220 |
| Cameroon | 2018 | 3,575 | 3,692 |
| Chad | 2015 | 6,283 | 6,439 |
| West Africa | | | |
| Benin | 2018 | 5,242 | 5,258 |
| Guinea | 2018 | 3,131 | 3,090 |
| Mali | 2018 | 3,875 | 4,090 |
| Senegal | 2017 | 4,251 | 3,906 |
| East Africa | | | |
| Burundi | 2017 | 5,006 | 5,157 |
| Ethiopia | 2016 | 3,810 | 3,993 |
| Rwanda | 2016 | 3,103 | 3,167 |
| Uganda | 2016 | 5,645 | 5,535 |
| Southern Africa | | | |
| Malawi | 2016 | 6,159 | 6,148 |
| Mozambique | 2015 | 2,044 | 2,223 |
| South Africa | 2016 | 1,301 | 1,308 |
| Zambia | 2018 | 3,699 | 3,623 |
| Zimbabwe | 2015 | 2,178 | 2,258 |
| Total | | 64,933 | 65,107 |

## Variables and measurement

**Outcome of interest.** The outcome variable of this study was HIV testing during pregnancy. Participants were asked if they tested for HIV as part of antenatal care and responses were dichotomous ("Yes" or "No").

**Explanatory variables.** Using the COM-B Model and HBM, we included both internal and external factors in our analysis. The COM-B model and HBM propose that knowledge and capability are important prerequisites for HIV testing. Capability can be psychological (knowledge) or physical (skills). Knowledge of the benefit of testing and the severity of not testing could influence women to test for HIV or not. As such, we included knowledge of MTCT in our analysis. Three main questions on the transmission of HIV from mother to child during pregnancy, delivery and breastfeeding were used to assess MTCT knowledge. Each respondent was given a score of 1 for each correct answer. The available scores ranged from 0–3. We categorised a score of "0" as low, a score of "1–2" as moderate, and a score of "3" as high. Also, according to COM-B model, opportunity and motivation are critical in factors that influence the uptake of HIV testing. Education, marital status and wealth status would not only influence an individual's capability to navigate and overcome the barriers to testing for HIV during pregnancy but also provide opportunity to access antenatal care and get tested. Age was categorised as "15–19 years", "20–24 years", "25–34 years" and "35–49 years". Marital status was classified as "Never married", "Currently married", "Previously married" and "Cohabiting". Education was measured by asking participants to report their highest level of education. Their responses were classified as "No formal education", "Primary education" and "Secondary and Higher education". Wealth status was assessed as an index that combines household assets and utilities. The index was then categorised as "Poor", "Middle" and "Rich". Also, we included health insurance coverage, which is a binary choice variable of Yes or No. All countries in the study sample had responses except for Senegal and Rwanda. The COM-B model emphasises the role of environmental factors in access and uptake of HIV testing. Where women live is an important factor that could determine if HIV testing is available and accessible. Health resources are unevenly distributed, and women in rural areas are disadvantaged in terms of access to health facilities that provide quality health care services. As such, we included residential area, which is divided into "Rural" and "Urban". Also, we included geopolitical zones, classified based on each country's setting.

According to the COM-B model, motivation is an important predictor of uptake of HIV testing. Even though the DHS did not directly measure women's motivation, other factors could influence one's motivation. We believe exposure to media programme on the importance of HIV testing could not only educate women on the need to test but only motivate them to test. Media exposure was constructed from three variables on the frequency of exposure to three different media outlets, which are print media, radio, and television using principal component analysis. The respondents were assigned 0 for "not at all", 1 for "less than once a week" and 2 for "at least once a week". These were added up across all respondents with overall scores of 0 to 6. We classified a score of "0"as "Low media exposure","1 to 3" as "Moderate media exposure", and 4 or greater as "High media exposure". This classification was applicable to all countries except Zambia. For Zambia, there was an additional category–"almost every day"– which we coded as 3. The available score for Zambia is 0 to 9. We classified "0" as "Low media exposure", "1 to 3" as "Moderate media exposure", and 4 or greater as "High media exposure".

We included countries as a variable to account for variation in the implementation of the "Opt-out" policy and the use of antenatal care services across the region. The "Opt-out" policy and rate of antenatal service utilisation are important factors influencing uptake of HIV testing during pregnancy. However, all women that did not receive antenatal care services also did

not test for HIV. As such, we are unable to include antenatal care attendance in our regression model. However, countries become an important proxy to account for the role of the disparate rate of antenatal care attendance across SSA counties. Also, the extent of implementation of the "Opt-out" policy, which is widely accepted and is implemented in SSA, varies. Therefore, countries become an important variable to account for these variations.

## Statistical analyses

The analyses were carried out using STATA Version 16.0. The analysis began by computing descriptive statistics, such as frequencies and percentages for the main explanatory variable and the outcome variable. We also performed Pearson's chi-square test analysis to examine the relationship between knowledge of MTCT and uptake of testing. We pooled the dataset of all countries to create a single dataset, yielding a total weighted sample of 65,106. Multivariable logistic regression models were used to examine factors associated with uptake of HIV testing during pregnancy in SSA. We stratified the regression analysis by countries to show the factors associated with prenatal testing in each country included. The results of the regression analyses were presented as odds ratios (OR), with their corresponding 95% confidence intervals (CI) signifying significance and precision of the reported OR.

## Ethical considerations

The DHS surveys are conducted after approval of ethical review bodies and authorisation by the country of the study. De-identified datasets are freely available on the DHS website (https://dhsprogram.com/data/available-datasets.cfm). Since this is a secondary analysis, we do not need to obtain a separate ethical approval other than those obtained when the primary data was collected.

## Results

### Descriptive results

We presented the prevalence of uptake of HIV testing during pregnancy in each of the 16 SSA countries included in the study in Fig 2. We found that the uptake of HIV testing ranges from about 6.1% in Chad to about 98.1% in Rwanda. We observed that the uptake of HIV testing was highest among women in Southern and Eastern African countries, whereas the uptake of HIV testing was lowest among women in Western and Central Africa countries (Fig 2).

Table 2A and 2B report the proportion of women who had HIV testing during pregnancy by background characteristics across the 16 SSA countries. Prevalence of prenatal HIV testing was higher among women older than 19 compared to women aged 19 or below in all countries except in Mozambique, South Africa and Zimbabwe. The proportion of never-married women who tested for HIV during pregnancy were higher compared to married women in Angola, Cameroon, Chad, Benin, Guinea, Mali, Ethiopia, Mozambique, Zambia and Zimbabwe. Also, the prevalence of HIV testing was highest among women who owned health insurance, resided in urban areas, had a high media exposure, belonged to rich wealth quintile and had secondary or higher level of education in all SSA countries studies. In all countries, uptake of HIV testing during pregnancy was significantly higher among women with moderate to high knowledge of MTCT than among those with low knowledge of MTCT. Even countries where uptake of HIV testing during pregnancy was high, women who had a high knowledge of MTCT were more likely to test for HIV compared to women who had low knowledge of MTCT (Rwanda 98.1% vs 87.1%, Malawi 91.7% vs 63.5%, South Africa 92.5% vs 60.9%, Uganda 92.4% vs 72.1%, Zambia 96.9% vs 54.9%, Zimbabwe 92.7% vs 46.6%).

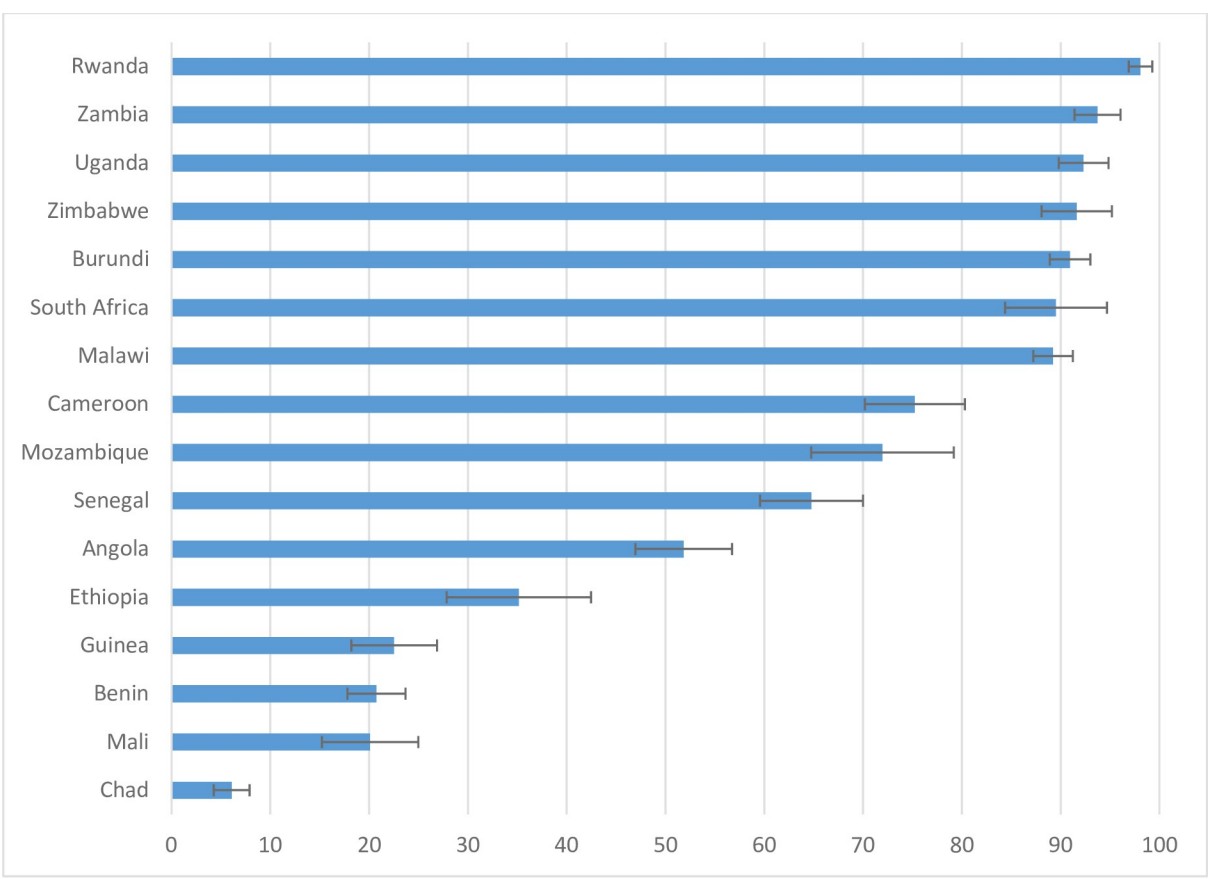

**Fig 2. Uptake of HIV testing during pregnancy across SSA countries.**

## Multivariable findings

We fitted two models to examine the factors associated with uptake of HIV testing during pregnancy. Model 1 was an unadjusted model with no covariates. Women aged 20–49 had a higher likelihood of prenatal uptake of HIV testing than women aged 15–19 years. Married women had lower odds of prenatal HIV testing relative to never-married women. Secondary education or higher, high wealth status, and high media exposure were associated with a higher likelihood of prenatal HIV testing. Ownership of health insurance was associated with higher odds of prenatal testing. In all countries, women who had moderate to high knowledge of MTCT had higher odds of HIV testing uptake during pregnancy compared with women who had low MTCT knowledge (Table 3).

Model 2 is the adjusted model where we included sociodemographic factors, media exposure, and health insurance as covariates. After controlling for countries in the adjusted model, women were more likely to test for HIV as part of antenatal care if they had a high knowledge of MTCT (AOR: 7.08; 95% CI: 6.66, 7.53), aged 35–49 (AOR: 1.33; 95% CI: 1.21, 1.46), had a secondary or higher level of education (AOR:2.57; 95% CI: 2.38, 2.77), belong to rich wealth quintile(AOR: 1.83; 95% CI: 1.70, 1.97), resided in the urban area (AOR: 1.68; 95% CI: 1.58, 1.80), exposed to the media (AOR: 1.77; 95% CI: 1.62, 1.94), and owned a health insurance cover (AOR: 1.41; 95% CI: 1.21, 1.64). We did not include ANC in the model, given that the prevalence of HIV testing uptake was 0% among women who did not receive ANC. However, we presented the result of testing prevalence and ANC attendance for all countries included,

**Table 2. Relationship between background factors and prenatal care uptake of HIV testing stratified by country.**

(A)

| Variables | SSA | Central Africa | | | West Africa | | | |
|---|---|---|---|---|---|---|---|---|
| | | Angola | Cameroon | Chad | Benin | Guinea | Mali | Senegal |
| **Age group in years** | | | | | | | | |
| 15–19 | 56.2 | 48.4 | 72.5 | 5.4 | 16.3 | 23.6 | 16.6 | 50.1 |
| 20–24 | 62.7 | 53.5 | 78.0 | 5.7 | 19.8 | 24.1 | 20.4 | 62.9 |
| 25–34 | 59.9 | 54.0 | 75.8 | 6.4 | 21.1 | 23.2 | 21.0 | 66.9 |
| 35–49 | 58.3 | 47.0 | 71.6 | 6.5 | 23.2 | 18.8 | 19.7 | 68.3 |
| **Marital Status** | | | | | | | | |
| Never Married | 75.3 | 54.8 | 87.6 | 11.4 | 30.5 | 34.0 | 23.7 | 62.7 |
| Currently Married | 54.9 | 42.7 | 70.5 | 5.7 | 19.2 | 21.6 | 20.0 | 64.8 |
| Previously | 70.4 | 52.1 | 73.3 | 8.3 | 18.2 | 39.3 | 19.8 | 64.5 |
| Cohabiting | 69.2 | 53.0 | 83.5 | 7.7 | 25.0 | 25.7 | 16.4 | 92.5 |
| **Education level** | | | | | | | | |
| None | 34.2 | 22.8 | 47.7 | 3.7 | 16.2 | 16.9 | 13.2 | 58.7 |
| Primary | 72.1 | 46.7 | 74.6 | 8.1 | 25.0 | 28.1 | 23.4 | 71.3 |
| Sec. & Higher | 81 | 82.7 | 93.2 | 14.1 | 31.7 | 48.1 | 46.5 | 77.4 |
| **Wealth Status** | | | | | | | | |
| Poor | 52.4 | 21.6 | 59.5 | 4.4 | 13.9 | 11.7 | 8.1 | 51.7 |
| Middle | 59.5 | 62.3 | 81.6 | 4.2 | 19.9 | 17.6 | 11.1 | 68.7 |
| Richest | 69.9 | 87.3 | 93.5 | 9.1 | 28.6 | 40.4 | 39.0 | 82.4 |
| **Residential Areas** | | | | | | | | |
| Rural | 56 | 21.4 | 65.2 | 4.3 | 17.2 | 14.0 | 13.0 | 56.9 |
| Urban | 70.2 | 72.1 | 88.7 | 13.7 | 26.4 | 44.2 | 46.8 | 79.9 |
| **Media Exposure** | | | | | | | | |
| Low | 48 | 23.0 | 58.5 | 4.1 | 14.4 | 12.7 | 8.5 | 41.3 |
| Moderate | 65.2 | 58.6 | 87.0 | 9.5 | 22.1 | 23.5 | 16.9 | 61.4 |
| High | 75.7 | 84.6 | 95.5 | 17.6 | 34.7 | 44.8 | 35.3 | 75.7 |
| **Health Insurance Cover** | | | | | | | | |
| No | 59 | 51.3 | 74.9 | 5.9 | 20.5 | 22.1 | 18.1 | |
| Yes | 85.7 | 66.7 | 96.6 | 90.6 | 50.3 | 71.4 | 63.1 | |
| **Know MTCT** | | | | | | | | |
| Low | 15.1 | 12.6 | 41.5 | 0.8 | 3.0 | 8.8 | 10.7 | 56.8 |
| Moderate | 74.5 | 61.3 | 74.8 | 20.4 | 52.7 | 27.9 | 31.2 | 66.7 |
| High | 76 | 74.8 | 81.2 | 26.3 | 49.3 | 32.1 | 23.1 | 68.5 |

(B)

| Variables | East Africa | | | | Southern Africa | | | | |
|---|---|---|---|---|---|---|---|---|---|
| | Burundi | Ethiopia | Rwanda | Uganda | Malawi | Mozambique | South Africa | Zambia | Zimbabwe |
| **Age group in years** | | | | | | | | | |
| 15–19 | 84.9 | 31.2 | 98.7 | 89.8 | 83.0 | 75.3 | 94.0 | 90.8 | 91.8 |
| 20–24 | 91.7 | 37.8 | 98.5 | 93.3 | 91.0 | 74.2 | 89.2 | 94.9 | 91.1 |
| 25–34 | 91.4 | 36.3 | 98.1 | 93.4 | 90.8 | 71.3 | 88.6 | 93.8 | 89.1 |
| 35–49 | 90.1 | 30.4 | 97.5 | 89.4 | 87.9 | 65.6 | 89.8 | 94.4 | 88.3 |
| **Marital Status** | | | | | | | | | |
| Never Married | 87.7 | 50.5 | 97.8 | 91.0 | 79.2 | 80.0 | 89.8 | 95.2 | 94.6 |
| Currently Married | 92.3 | 35.3 | 98.9 | 92.5 | 90.4 | 67.0 | 90.3 | 93.7 | 89.5 |
| Previously | 90.7 | 27.7 | 95.7 | 91.7 | 86.3 | 66.6 | 97.4 | 91.0 | 89.3 |

(*Continued*)

| | | | | | | | | | |
|---|---|---|---|---|---|---|---|---|---|
| Cohabiting | 87.8 | 39.1 | 97.5 | 92.5 | 88.7 | 79.1 | 86.0 | 97.4 | 93.1 |
| Education level | | | | | | | | | |
| None | 89.7 | 24.0 | 95.4 | 88.3 | 86.3 | 58.3 | 93.6 | 80.5 | 87.3 |
| Primary | 91.4 | 43.5 | 98.4 | 90.4 | 88.6 | 73.2 | 76.8 | 93.3 | 83.2 |
| Sec. & Higher | 94.0 | 82.2 | 98.9 | 97.6 | 93.0 | 91.0 | 90.5 | 97.5 | 93.1 |
| Wealth Status | | | | | | | | | |
| Poor | 88.7 | 20.5 | 97.3 | 89.3 | 87.9 | 61.2 | 87.1 | 91.0 | 86.9 |
| Middle | 91.9 | 30.3 | 98.8 | 92.6 | 89.9 | 69.4 | 93.7 | 93.9 | 90.3 |
| Richest | 93.2 | 57.8 | 98.7 | 95.8 | 90.8 | 89.0 | 90.3 | 97.8 | 93.4 |
| Residential Areas | | | | | | | | | |
| Rural | 90.4 | 29.1 | 98.0 | 91.1 | 88.7 | 67.1 | 92.4 | 91.9 | 88.7 |
| Urban | 96.6 | 79.1 | 98.6 | 97.0 | 92.4 | 86.2 | 87.9 | 97.5 | 93.0 |
| Media Exposure | | | | | | | | | |
| Low | 88.6 | 26.0 | 96.3 | 87.8 | 87.4 | 65.4 | 77.8 | 92.0 | 85.2 |
| Moderate | 93.5 | 48.5 | 98.2 | 93.0 | 90.6 | 76.5 | 91.2 | 94.3 | 90.3 |
| High | 96.0 | 75.2 | 99.4 | 96.0 | 92.5 | 93.4 | 91.1 | 97.3 | 95.6 |
| Health Insurance Cover | | | | | | | | | |
| No | 90.5 | 34.1 | | 92.3 | 89.2 | 71.8 | 89.8 | 93.7 | 89.3 |
| Yes | 92.4 | 63.1 | | 97.0 | 92.3 | 89.8 | 84.1 | 97.9 | 97.6 |
| Know MTCT | | | | | | | | | |
| Low | 25.5 | 12.9 | 87.1 | 74.1 | 63.5 | 56.4 | 60.9 | 54.9 | 46.6 |
| Moderate | 91.6 | 35.0 | 98.3 | 93.4 | 90.5 | 75.1 | 95.1 | 96.4 | 86.1 |
| High | 94.2 | 44.7 | 98.1 | 92.4 | 91.7 | 77.2 | 92.5 | 96.9 | 92.7 |

and the results show that testing the proportion of women who tested during pregnancy was higher in countries with HIV coverage of ANC than in those with low coverage.

We stratified the results by country to report on the homogeneity and heterogeneity of the results. The findings are shown in S1–S6 Tables. In both the adjusted and unadjusted models, the odds of uptake of HIV testing as part of antenatal care was higher among women with high MTCT knowledge in all countries studied.

Older age (aged 20 to 49) was significantly associated with a higher likelihood of uptake of HIV testing during pregnancy only in Angola, Benin, Mali, Senegal, and Burundi. There was heterogeneity in the result of the association between marital status and uptake of HIV testing. In most countries, marital status was not significantly related to prenatal uptake of testing. Married women had higher odds of uptake of HIV testing in Burundi compared to never-married women. However, the contrast was found in Zimbabwe, where married women were 63% less likely to test during antenatal care. In all countries studied, except Malawi and South Africa, women who had a secondary or higher level of education were more likely to test for HIV as part of ANC compared to women who had no formal education. Rich wealth status was significantly associated with higher odds of uptake of HIV testing in all countries, except Malawi, Zimbabwe, Zambia, South Africa, Uganda, Rwanda, and Chad.

Similarly, urban-dwelling was associated with a higher likelihood of HIV testing uptake in all countries, except in Zambia, Zimbabwe, South Africa, Uganda and Rwanda. Surprisingly, women were significantly less likely to test for HIV during ANC if they reside in urban areas in South Africa. High media exposure was associated with a higher odd of prenatal care uptake of HIV testing in all countries studied, except in Rwanda, Uganda, and South Africa. Lastly, ownership of health insurance was significantly associated with uptake of HIV testing as part

**Table 3. Multivariable models showing factors associated with uptake of HIV testing in SSA.**

| Variables | Uptake of HIV testing during pregnancy | |
|---|---|---|
| | UOR [95% CI] | AOR [95% CI] |
| Age group in years | | |
| 15–19 | Ref | Ref |
| 20–24 | 1.31 [1.24,1.38]*** | 1.21 [1.11,1.31]*** |
| 25–34 | 1.15 [1.09,1.21]*** | 1.30 [1.20,1.41]*** |
| 35–49 | 1.07 [1.01,1.13]* | 1.33 [1.21,1.46]*** |
| Marital Status | | |
| Never Married | Ref | Ref |
| Currently married | 0.42 [0.39,0.44]*** | 0.98 [0.88,1.09] |
| Previously married | 0.86 [0.78,0.94]** | 0.94 [0.82,1.07] |
| Cohabiting | 0.79 [0.73,0.85]*** | 1.08 [0.97,1.21] |
| Education level | | |
| None | Ref | Ref |
| Primary | 5.67 [5.45,5.89]*** | 1.60 [1.51,1.70]*** |
| Secondary & Higher | 8.84 [8.43,9.28]*** | 2.57 [2.38,2.77]*** |
| Wealth Status | | |
| Poor | Ref | Ref |
| Middle | 1.37 [1.31,1.43]*** | 1.37 [1.29,1.46]*** |
| Rich | 2.04 [1.97,2.11]*** | 1.83 [1.70,1.97]*** |
| Residence | | |
| Rural | Ref | Ref |
| Urban | 1.69 [1.63,1.75]*** | 1.68 [1.58,1.80]*** |
| Media Exposure | | |
| Low | Ref | Ref |
| Moderate | 2.12 [2.05,2.19]*** | 1.43 [1.35,1.51]*** |
| High | 3.30 [3.14,3.48]*** | 1.77 [1.62,1.94]*** |
| Health Insurance Cover | | |
| No | Ref | Ref |
| Yes | 3.96 [3.54,4.43]*** | 1.41 [1.21,1.64]*** |
| Knowledge of MTCT | | |
| Low | Ref | Ref |
| Moderate | 19.46 [18.33,20.65]*** | 5.92 [5.49,6.38]*** |
| High | 20.29 [19.32,21.32]*** | 7.08 [6.66,7.53]*** |
| Countries | | |
| Angola | Ref | Ref |
| Cameroon | 3.47 [3.17,3.81]*** | 2.84 [2.53,3.19]*** |
| Chad | 0.06 [0.06,0.07]*** | 0.14 [0.12,0.17]*** |
| Benin | 0.29 [0.26,0.31]*** | 0.40 [0.36,0.45]*** |
| Guinea | 0.30 [0.27,0.33]*** | 0.32 [0.28,0.37]*** |
| Mali | 0.26 [0.24,0.28]*** | 0.22 [0.20,0.25]*** |
| Senegal | 1.63 [1.50,1.77]*** | 2.13 [1.90,2.39]*** |
| Burundi | 10.71 [9.60,11.94]*** | 10.16 [8.92,11.58]*** |
| Ethiopia | 0.73 [0.67,0.80]*** | 0.91 [0.81,1.03] |
| Rwanda | 56.17 [43.18,73.05]*** | 36.38 [27.74,47.71]*** |
| Uganda | 13.58 [12.13,15.19]*** | 10.14 [8.92,11.54]*** |
| Malawi | 9.73 [8.82,10.73]*** | 9.38 [8.28,10.62]*** |
| Mozambique | 3.44 [3.07,3.86]*** | 4.15 [3.61,4.77]*** |

*(Continued)*

**Table 3.** (Continued)

| Variables | Uptake of HIV testing during pregnancy | |
|---|---|---|
| | UOR [95% CI] | AOR [95% CI] |
| South Africa | 10.81 [8.89,13.15]*** | 5.58 [4.47,6.97]*** |
| Zambia | 15.48 [13.46,17.81]*** | 15.72 [13.36,18.48]*** |
| Zimbabwe | 11.07 [9.47,12.94]*** | 5.88 [4.92,7.02]*** |

AOR is the adjusted odds ratio, UOR is the unadjusted odds ratio, ref is the reference; Exponentiated coefficients; 95% confidence intervals in brackets.

* $p < 0.05$

** $p < 0.01$

*** $p < 0.001$.

of ANC in all countries, except in Benin, Burundi, Uganda, South Africa, Zambia, Zimbabwe, Malawi, and Mozambique.

## Discussion

We examined the coverage and factors associated with HIV testing among pregnant women in SSA. Our study is timely, considering the global need for evidence to support the achievement of the UNAIDS 95-95-95 targets [31]. Understanding the gaps in coverage of HIV testing during pregnancy is critical for designing programmes to address the gaps and ensure even progress in SSA [2, 32]. Our analysis showed uneven progress in expanding access to HIV testing during pregnancy in SSA. East and Southern African countries like Rwanda, Malawi, Zimbabwe and South Africa have expanded access to prenatal HIV testing for most women, irrespective of their education level, wealth status and place of residence. In contrast, prenatal care HIV testing coverage was low in all West, and Central African countries studied. We also found a higher uptake of HIV testing among women in Southern and Eastern African countries, compared to Western and Central African countries.

There are several pathways to understand these findings. First, the Southern and Eastern African regions are most affected by HIV in the world (S7 Table) and are home to the largest number of people living with HIV (20.6 million) [1]. In line with this, several countries in the region such as Botswana, Kenya, Uganda, Malawi, and Rwanda have not only implemented national campaigns to encourage uptake of HIV testing and counselling (HTC), they have also implemented effective and efficient PMTCT, ensuring that most pregnant women are tested for HIV and those diagnosed are placed on treatment [33–35]. They have deployed community-based testing, which supports provider-initiated testing. Also, workplace and door-to-door testing and self-testing, using rapid diagnostic tests, are being implemented in these sub-regions [9].

Due to the heavy burden of HIV in the Eastern and Southern Africa regions, attention and efforts of global developmental partners are concentrated on the region. Also, the governments of countries in these regions, particularly South Africa, have invested heavily towards reducing new HIV transmission and eliminating AIDS. The partnership of the local and global effort in ending AIDS and preventing HIV transmission have no doubt bear tremendous results in East and Southern Africa, including expanded access to HIV testing for pregnant women and reduced MTCT of HIV. It is therefore imperative to focus more attention on West and Central African countries to replicate the results recorded in East and Southern African in their region.

Apart from these explanations, the high knowledge of MTCT and high HIV testing among women in the Southern and Eastern African regions could also be linked to the high uptake of

antenatal care services (see S7 Table). Women in Central and West Africa were, on average, less likely to receive antenatal care compared to women in Southern and Eastern Africa (see S7 Table). Nevertheless, despite the remarkable progress, particularly in South Africa, Rwanda, and Uganda and Zimbabwe, no country has universal testing of pregnant women. As such, programme implementers must make an effort to reach the missing pregnant women in the HIV care cascade.

Also consistent with previous research [2, 34, 36] women who had secondary or higher education, who owned health insurance and who resided in the wealthiest households were more likely to be tested for HIV during pregnancy compared to those who did not have formal education, did not have health insurance and lived in the poorest households. This finding was significant in most countries studied, indicating socioeconomic gaps in ANC coverage of HIV testing. Only a few countries in SSA (Rwanda, South Africa, Zimbabwe, Malawi and Zambia) have managed to eliminate wealth and education inequality in access to HIV testing for pregnant women. These countries have demonstrated that eliminating inequalities in access to prenatal HIV testing is possible with the implementation of equitable policies. As such, other countries in the region could draw lessons from these countries to provide universal access to care and ensure that no one is left behind.

In most SSA countries studied, we observed rural-urban disparity in coverage of HIV testing during pregnancy, highlighting the need to scale up prenatal HIV coverage in rural areas. Rwanda, Zambia, Zimbabwe, Malawi, and Mozambique are the few countries where the rural and urban disparity in coverage of prenatal HIV testing has been eliminated. Equitable access to prenatal testing is vital for the health of women and babies in SSA. West and Central African countries must address the rural and urban disparity in coverage of prenatal HIV testing to save the lives of mothers and babies. Scaling up access to HIV testing in rural areas alone will remarkably increase uptake in the West and Central Africa. Our findings on the association between marital status and prenatal care uptake of HIV testing suggest no significant association in 14 of the 16 countries studied. Married women were more likely to test for HIV in Burundi but less likely to test in Zimbabwe. More studies are needed to understand this result.

The findings from this study have implications for policy and practice regarding the prevention of MTCT in SSA. As shown in S7 Table, new HIV infections are high in Central and West African countries, and these countries also have lower uptake of antenatal care, limiting the level of prenatal HIV screening. Efforts to reduce or eliminate MTCT should focus on West and Central Africa. Such efforts must target women who do not seek antenatal care services. Our findings also underscore the need for PMTCT programmes to pay particular attention to knowledge on MTCT and socioeconomic inequality. Community-based interventions delivered through women and church leaders could help improve women's MTCT knowledge in underserved settings, resulting in increased knowledge and uptake of HIV testing.

## Strength and limitations

The strength of the study lies in the use of nationally representative survey data and the large sample size. Notwithstanding, the study is without limitations. First, the use of the cross-sectional study design, as employed in the DHS, limits the capacity of the authors to attribute causality to the findings. Again, the study adopted self-reporting of past events (prenatal care uptake of HIV testing), which is subjected to social desirability and recall bias. The DHS, however, limits responses to this question to two years to reduce the impact of recall bias. Lastly, there are several other unmeasured confounders such as couple testing, availability of testing services and how testing is done, which could potentially influence the uptake of HIV testing during ANC.

## Conclusion

The findings of the present study highlight the between countries and sub-regional disparities in prenatal care uptake of HIV testing in SSA countries. Even though no country has a universal coverage of HIV testing of pregnant women, East and Southern African countries have made remarkable progress towards ensuring no pregnant woman is not tested. However, the coverage of testing is incredibly low in West and Central Africa, with the rich and well educated having better access to testing, while the poor are rarely tested. Addressing the inequitable access and coverage of HIV testing among pregnant women is vital in these sub-regions. In all countries studied, knowledge of MTCT was linked with a higher likelihood of testing, which calls for appropriate interventions to increase awareness of MTCT. Such interventions could be delivered through community leaders and mass media. In doing this, priority should also be given to pregnant women in underserved settings, especially in West and Central Africa.

## Supporting information

**S1 Table. Adjusted and unadjusted logistic regression models showing factors associated with prenatal uptake of HIV testing in Angola, Cameroun and Chad.**
(DOCX)

**S2 Table. Adjusted and unadjusted logistic regression models showing factors associated with prenatal uptake of HIV testing in Benin, Guinea and Mali.**
(DOCX)

**S3 Table. Adjusted and unadjusted logistic regression models showing factors associated with prenatal uptake of HIV testing in Senegal, Burundi and Ethiopia.**
(DOCX)

**S4 Table. Adjusted and unadjusted logistic regression models showing factors associated with prenatal uptake of HIV testing in Rwanda, Uganda and South Africa.**
(DOCX)

**S5 Table. Adjusted and unadjusted logistic regression models showing factors associated with prenatal uptake of HIV testing in Zambia, Zimbabwe and Malawi.**
(DOCX)

**S6 Table. Adjusted and unadjusted logistic regression models showing factors associated with prenatal uptake of HIV testing in Mozambique.**
(DOCX)

**S7 Table. Antenatal care, HIV prevalence and new HIV infections.**
(DOCX)

## Acknowledgments

We acknowledge the DHS program for making the datasets available for our use.

## Author Contributions

**Conceptualization:** Oluwafemi Emmanuel Awopegba, Anthony Idowu Ajayi.

**Data curation:** Oluwafemi Emmanuel Awopegba, Anthony Idowu Ajayi.

**Formal analysis:** Oluwafemi Emmanuel Awopegba.

**Investigation:** Anthony Idowu Ajayi.

**Methodology:** Oluwafemi Emmanuel Awopegba, Anthony Idowu Ajayi.

**Supervision:** Anthony Idowu Ajayi.

**Validation:** Anthony Idowu Ajayi.

**Writing – original draft:** Oluwafemi Emmanuel Awopegba, Amarachi Kalu, Bright Opoku Ahinkorah, Abdul-Aziz Seidu, Anthony Idowu Ajayi.

**Writing – review & editing:** Oluwafemi Emmanuel Awopegba, Amarachi Kalu, Bright Opoku Ahinkorah, Abdul-Aziz Seidu, Anthony Idowu Ajayi.

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
