## [Decision Letter · Decision Letter 0]

28 Aug 2020

PONE-D-20-21800

Knowledge of mother-to-child transmission of HIV and uptake of HIV testing during pregnancy: findings from 14 sub-Saharan African countries

PLOS ONE

Dear Dr. Ajayi,

Thank you for submitting your manuscript to PLOS ONE. After careful consideration, we feel that it has merit but does not fully meet PLOS ONE’s publication criteria as it currently stands. Therefore, we invite you to submit a revised version of the manuscript that addresses the points raised during the review process.

Many thanks for your submission which highlights important discrepancies in and consequences of vertical transmission of HIV policies across the region. Both reviewers have highlighted concerns regarding the presentation of the methodology and the conclusions drawn. The premise that there is a straight-forward causal relationship between knowledge and testing needs to be examined in more detail to account for study design, confounding and colinearity. Here regional PMTCT policy and practice are important: education and counselling as part of PMTCT, opt-out testing versus opt-in etc. Similarly,as noted in the discussion, ANC attendance is vital. 

Please check the formatting of the tables including denominators (e.g. Table 2), units (e.g. Table 3) and general formatting (e.g. Table 5) where appropriate.

Please note, one set of comments is included as an attachment.

We look forward to receiving your revised manuscript.

Kind regards,

Emma K. Kalk

Academic Editor

PLOS ONE

Journal Requirements:

2. In your Methods section, please provide additional information about the demographic details of your participants. Please ensure you have provided sufficient details to replicate the analyses such as:

a)  a description of any inclusion/exclusion criteria that were applied to participant inclusion in the analysis (specifying what illnesses' were considered),

b) a table of relevant demographic details.

4. We note you have included tables to which you do not refer in the text of your manuscript. Please ensure that you refer to Tables 1 and 2 in your text; if accepted, production will need this reference to link the reader to each Table.

Reviewers' comments:

Reviewer's Responses to Questions

**Comments to the Author**

1. Is the manuscript technically sound, and do the data support the conclusions?

Reviewer #1: Partly

Reviewer #2: Partly

2. Has the statistical analysis been performed appropriately and rigorously? 

Reviewer #1: Yes

Reviewer #2: No

3. Have the authors made all data underlying the findings in their manuscript fully available?

Reviewer #1: Yes

Reviewer #2: Yes

4. Is the manuscript presented in an intelligible fashion and written in standard English?

Reviewer #1: Yes

Reviewer #2: Yes

5. Review Comments to the Author

Reviewer #1: Knowledge of mother-to-child transmission of HIV and uptake of HIV testing during

pregnancy: findings from 14 sub-Saharan African countries

Awopegba OE, et al

Comments

Abstract

Line 49-51:‘The odds of prenatal care HIV testing was significantly higher [21.21, CI=20.03,22.46] in countries where most women knew of MTCT compared to countries where most women did not know of MTCT routes of transmission.’

Comment

While from the these results are true based on the data used, but there some confounding factors, for example in Uganda it is kind of a must for every pregnant mother who attends ANC to test for HIV, rather than a voluntary choice of the mother. So these health systems differences across the countries have to be taken into account, before we conclude that the testing rates of HIV during prenatal are due to the level of knowledge. Like I have mentioned in these countries like Rwanda and Uganda, HIV testing is ‘mandatory’ during ANC, and the mothers are also given health education. One would love to know the health systems dynamics for PMTCT in West Africa, where testing and knowledge are low, so as to put these findings into context.

Line 53-56: The conclusion of ‘educating women of the risk of MTCT could be an important strategy for increasing HIV testing uptake, especially in West and Central Africa, where the rate of testing during pregnancy remains low, and the rate of MTCT remain high, is not the only solution, when other factors are not addressed, because knowledge alone without addressing health systems barriers is not enough.

Line 77-78.’ Global north countries with a universal prenatal care HIV screening among pregnant women have nearly eliminated MTCT of HIV.’ Need to provide the source of this information. But also this statement underscores the importance of having the prenatal care HIV screening among pregnant women. It is not clear if these services are available in West Africa, where MTCT HIV rates are high and testing is low.

Line 89-91: The authors emphasize the importance of knowledge in facilitation HIV testing among prenatal mothers, but literature has shown that knowledge alone is not enough.

‘Knowledge of MTCT is considered essential to facilitate HIV testing since women with knowledge of HIV transmission have a better understanding and appreciation of the need for HIV testing and the perceived benefit of testing not only for them but also for their infant 17’.

Line 92-107: The authors are using The Health Belief Model (HBM) with the premise that knowledge is enough for mothers to test. I feel they should have used more comprehensive models of behaviour change e.g. the approach the Capability, Opportunity and Motivation Model of Behaviour (COM-B) model (Michie S 2011), see figure below. According to the model, behaviour is a product of three necessary conditions; capability, opportunity, and motivation. Capability can be psychological (knowledge) or physical (skills), opportunity can be social (societal influences) or physical (environmental resources) while motivation can be automatic (emotion) or reflective (beliefs, intentions). Such model like COM-B gives a comprehensive analysis of the issues that affect prenatal HIV testing. Therefore, the current analysis gives a narrow picture.

Figure 1. COM-B Model Michie S 2011.

Therefore, the authors need to recognize the limitations of their theoretical approach, is not comprehensive enough in exploring the factors that affect prenatal HIV testing. But also the secondary data limitations, not able to have collected other data such as health systems factors e.g availability of testing services, how testing is done, etc.

Line 210-2013: Table 3. The authors don’t give the reference point and also should provide the confidence intervals.

Line 2017-2025: In all countries, women who had moderate to high knowledge of MTCT had higher odds of HIV testing uptake during pregnancy compared with women who had low MTCT 219 knowledge. In Model 2, we added sociodemographic factors, media exposure, and health insurance as covariates. After adjusting for covariates, the magnitude and direction of effect persisted, indicating a strong and robust effect of MTCT knowledge on uptake of HIV testing during pregnancy. The odds of prenatal HIV testing uptake was higher among women with high MTCT 223 knowledge, especially in Chad (AOR: 38.30; 95% CI: 26.23, 55.93), Benin (AOR: 40.65; 95% CI: 224 31.95, 51.72), Angola (AOR:8.14; 95% CI:6.77, 9.78), Burundi (AOR: 53.37; 95% CI: 36.00, 225 79.11), and Zimbabwe (AOR: 19.1; 95% CI: 10.4, 35.0).

Comment: Though this data analysis is correct, it has limitation of health systems issues on the ground.

Table 4; Also does not provide reference points/measurements.

Comment: It would also be important for this study to give us a clue on the prevalence of couple testing, given that the pregnant mother testing alone without the partner leaves a gap in terms of achieving the PMTCT targets.

The authors should acknowledge the limitations of this secondary data analysis.

Reviewer #2: Review attached separately .

6. PLOS authors have the option to publish the peer review history of their article (what does this mean?). If published, this will include your full peer review and any attached files.

Reviewer #1: No

Reviewer #2: No

---

## [Author Response · Author response to Decision Letter 0]

8 Oct 2020

PlosOne paper review: PONE-D-20-21800 – 

Title of paper: Knowledge of mother-to-child transmission of HIV and uptake of HIV testing during pregnancy: findings from 14 sub-Saharan African countries

Dear Editor:

I am pleased to re-submit the revised version of our manuscript. We appreciate the time taken to critically review this paper and the constructive comments we received from you and the anonymous reviewers. We believe that addressing all of your concerns as well as theirs, has contributed significantly to our paper. We have considered all the comments and have addressed each of the concerns as outlined below. 

Editors comments

Many thanks for your submission which highlights important discrepancies in and consequences of vertical transmission of HIV policies across the region. Both reviewers have highlighted concerns regarding the presentation of the methodology and the conclusions drawn. The premise that there is a straight-forward causal relationship between knowledge and testing needs to be examined in more detail to account for study design, confounding and collinearity. Here regional PMTCT policy and practice are important: education and counselling as part of PMTCT, opt-out testing versus opt-in etc. Similarly, as noted in the discussion, ANC attendance is vital. 

Response: We thank the editor and the reviewers for all these insightful comments. We have addressed these comments, making changes to the methods, results, and addressing the issues of collinearity and regional PMTCT policy and practice. We have specifically emphasized why ANC attendance is critical to the uptake of HIV testing during pregnancy. Also, our country stratified analysis suggests that knowledge of MTCT is important for HIV testing uptake given this variable was significant for all countries included in the analysis. We have now focused on all factors associated with uptake of HIV testing during pregnancy. 

Comment

Please check the formatting of the tables, including denominators (e.g. Table 2), units (e.g. Table 3) and general formatting (e.g. Table 5) where appropriate.

Response: We have made these corrections to the tables. Many thanks for suggesting these corrections.

Reviewer 1

Comments

Reviewer #1: Knowledge of mother-to-child transmission of HIV and uptake of HIV testing during pregnancy: findings from 14 sub-Saharan African countries

Awopegba OE, et al

Comments

Abstract

Line 49-51:‘The odds of prenatal care HIV testing was significantly higher [21.21, CI=20.03,22.46] in countries where most women knew of MTCT compared to countries where most women did not know of MTCT routes of transmission.’

Comment

While from the these results are true based on the data used, but there some confounding factors, for example in Uganda it is kind of a must for every pregnant mother who attends ANC to test for HIV, rather than a voluntary choice of the mother. So these health systems differences across the countries have to be taken into account, before we conclude that the testing rates of HIV during prenatal are due to the level of knowledge. Like I have mentioned in these countries like Rwanda and Uganda, HIV testing is ‘mandatory’ during ANC, and the mothers are also given health education. One would love to know the health systems dynamics for PMTCT in West Africa, where testing and knowledge are low, so as to put these findings into context.

Line 53-56: The conclusion of ‘educating women of the risk of MTCT could be an important strategy for increasing HIV testing uptake, especially in West and Central Africa, where the rate of testing during pregnancy remains low, and the rate of MTCT remain high, is not the only solution, when other factors are not addressed, because knowledge alone without addressing health systems barriers is not enough.

Response: We thank the reviewer for these comments. We have used this insight to improve our introduction and discussion. Also, we have revised the conclusion to highlight the role of other factors in our findings.

Comment

Line 77-78.’ Global north countries with a universal prenatal care HIV screening among pregnant women have nearly eliminated MTCT of HIV.’ Need to provide the source of this information. But also this statement underscores the importance of having prenatal care HIV screening among pregnant women. It is not clear if these services are available in West Africa, where MTCT HIV rates are high and testing is low.

Response: We have provided a reference to this sentence. HIV testing is generally available in all West African countries; however, the challenge remains to ensure women access prenatal care and also ensure all women who do are tested as part of prenatal care. In Nigeria, for example, not all women who received prenatal care are tested. As such, significant gaps remain in terms of universal testing of women who present for antenatal care. Nevertheless, the larger percentage of unreached women remains those who never received prenatal care. 

Comment

Line 89-91: The authors emphasize the importance of knowledge in facilitation HIV testing among prenatal mothers, but literature has shown that knowledge alone is not enough.

‘Knowledge of MTCT is considered essential to facilitate HIV testing since women with knowledge of HIV transmission have a better understanding and appreciation of the need for HIV testing and the perceived benefit of testing not only for them but also for their infant 17’.

Response: We agree with this comment and have revised the manuscript accordingly. We have highlighted the role of PMTCT policies, health system strengthening, and addressing of demand and supply factors in improving coverage of prenatal care testing. 

Comment

Line 92-107: The authors are using The Health Belief Model (HBM) with the premise that knowledge is enough for mothers to test. I feel they should have used more comprehensive models of behaviour change e.g. the approach the Capability, Opportunity and Motivation Model of Behaviour (COM-B) model (Michie S 2011), see figure below. According to the model, behaviour is a product of three necessary conditions; capability, opportunity, and motivation. Capability can be psychological (knowledge) or physical (skills), opportunity can be social (societal influences) or physical (environmental resources) while motivation can be automatic (emotion) or reflective (beliefs, intentions). Such model like COM-B gives a comprehensive analysis of the issues that affect prenatal HIV testing. Therefore, the current analysis gives a narrow picture.

Figure 1. COM-B Model Michie S 2011.

Therefore, the authors need to recognize the limitations of their theoretical approach, is not comprehensive enough in exploring the factors that affect prenatal HIV testing. But also the secondary data limitations, not able to have collected other data such as health systems factors e.g availability of testing services, how testing is done, etc.Response: We appreciate the reviewer for not only providing comments but also suggesting a theory. We have read this theory and have applied it in this study. The theory provides a comprehensive analysis of barriers and enablers of HIV testing and we have related the constructs of the theory to our study. 

Line 210-2013: Table 3. The authors don’t give the reference point and also should provide the confidence intervals.

Response: Table 3 is shows the cross-tabulated results of knowledge and prenatal care testing from a chi-square test of independence. Hence, it does not require confidence intervals intervals nor reference point as required in a binary logistic regression table. 

Line 2017-2025: In all countries, women who had moderate to high knowledge of MTCT had higher odds of HIV testing uptake during pregnancy compared with women who had low MTCT 219 knowledge. In Model 2, we added sociodemographic factors, media exposure, and health insurance as covariates. After adjusting for covariates, the magnitude and direction of effect persisted, indicating a strong and robust effect of MTCT knowledge on uptake of HIV testing during pregnancy. The odds of prenatal HIV testing uptake was higher among women with high MTCT 223 knowledge, especially in Chad (AOR: 38.30; 95% CI: 26.23, 55.93), Benin (AOR: 40.65; 95% CI: 224 31.95, 51.72), Angola (AOR:8.14; 95% CI:6.77, 9.78), Burundi (AOR: 53.37; 95% CI: 36.00, 225 79.11), and Zimbabwe (AOR: 19.1; 95% CI: 10.4, 35.0).

Comment: Though this data analysis is correct, it has a limitation of health systems issues on the ground.

Response: We have highlighted this limitation under the strengths and limitations of our study. 

Table 4; Also does not provide reference points/measurements.

Comment: It would also be important for this study to give us a clue on the prevalence of couple testing, given that the pregnant mother testing alone without the partner leaves a gap in terms of achieving the PMTCT targets.

The authors should acknowledge the limitations of this secondary data analysis.

Response: The reference point in Table 4 is denoted by “Ref”. In relation to couple testing, we have acknowledged this as a limitation since the variable was not available in the datasets.

Review 2 

Comments:

- The DHS is a cross-sectional study, which means you can’t draw directional associations, or cause-and-effect. Therefore, you cannot say that education caused testing (direction of effect), because it’s just as plausible that testing access resulted in greater knowledge of MTCT, especially since education is usually part of the pre-test process in ANC. To be able to draw the conclusion you’ve drawn, you’d have to have some indication that women had been educated on MTCT prior to their test, something hard to do in a cross-sectional study if there wasn’t a question specifically targeting when the person gained knowledge about PMTCT

Response: We did indicate this under our study limitation.

- There’s no reflection in your methods, results or discussion about the possibility of collinearity and confounding variables (ie. media exposure and knowledge) – this is a gap that needs to be addressed

Response: We have now addressed this. 

- Your main independent variable and your dependent variable are almost certainly highly correlated. I’d suggest that this is because education on PMTCT happens in ANC as part of the pre-test counseling process, so knowledge=testing. This is really important and isn’t addressed anywhere in your paper or accounted for in your analysis plan. 

Response: The correlation coefficient was approximately 0.5. We agree that this is high and have discussed it. 

- I think your discussion needs to dig a little deeper into what your results are telling you:

o It’s clear that media exposure is associated with testing, which implies that the knowledge influencing behavior (testing) comes from somewhere other than the ANC setting. But the analysis doesn’t account for this, and there’s no mention of this in the discussion

- Response: We have included this in our discussion. 

- 

o Looking at marital status and testing – the relationship is the inverse of what you’d want to see; marriage/cohabitating is associated with not testing – other national PMTCT evaluations from SSA show similar findings and it’s worth discussing this in your paper because it appears to hold true across multiple countries in your analysis 

- Response: We have discussed the effect of marital status. However, we found a significant association between marital status and tesing only in 2 of the 16 countries. 

- In the intro section more work needs to be done to highlight the major variation in PMTCT success within Africa. Several countries have demonstrated successful MTCT at the levels seen in developed nations (see Zim, SA and Malawi publications), but this doesn’t come out clearly when all grouped together as ‘PMTCT programmes in SSA’.

Response: We have specifically referenced these countries in the introduction. We have further highlighted the remarkable progress made particularly by southern African countries, including Zim, SA and Malawi. 

- Media exposure classification for Zambia needs to be revised – this appears to be inconsistent with classification for all other countries – if ‘almost every day’, then this should be grouped with the high media exposure, not moderate media exposure

Response: We have grouped ‘almost every day’as high media exposure.

- It’s unclear from the description in the methods whether weighting was done correctly during analysis. Weight variables are usually provided with DHS datasets and the use of these in your analysis should be described in your methods, and your results presented as ‘weighted’

Response: Women’s sample weights were appropriately applied to obtain unbiased estimates according to the DHS guidelines. We have made this clear in the methods.

- Table content and formatting: The table selection, content and formatting needs to be worked on. Suggest reviewing other PMTCT papers with regression analyses and identifying the ‘typical’ flow of tables included in the results sections.

Response: We have revised the tables and formatted them appropriately. 

Specific comments:

Abstract

Line 34 – ‘inefficient’ isn’t a good word here. Please identify a better descriptor: ie. ‘Incomplete’? ‘Inadequate coverage of’? ‘inadequate uptake’?

Response: We thank the reviewer for this correction. We have effected the change. 

Line 38 – what software was used?

Response: We used Stata version 16 and have indicated this at the statistical analyses section. 

Introduction

Line 64-66 – sentences 1 and 2 are repetitive. Combine into a single sentence.

Response: Done

Line 77 – update your literature review to include National PMTCT evaluations which have been published since 2015 from Malawi, Zimbabwe and South Africa – all have demonstrated in nationally representative studies that MTCT at 6-12wks postpartum is comparable to ‘global north’. Suggest you focus the intro section on the wide variation of PMTCT success within Africa for greater impact to the reader. 

Response: We thank the reviewer for the positive feedback. We have focused on introduction on the wide variation in PMTCT success within Africa

Line 80-82 – revise sentence based on guidance provided in previous comment

Response: We thank the reviewer for important suggestions. We have revised the sentence. 

Line 86 – not clear what ‘operate through knowledge’ means – revise sentence

Response: deleted

Line 94 – please provide reference(s) for first sentence in the paragraph

Response: Done

Line 102 – remove ‘in the main’. Suggest starting sentence with “we theorized that”

Response: We have deleted the sentence

Lines 108-118 – move to methods section, and remove the sentences about how the findings can be used (those below in your conclusions section if still relevant after doing the analysis)

Response: done

Methods

Line 120 – ‘uses’ should be ‘used’ – make sure you use past tense consistently throughout the paper (ie see lines 124, 125 and 126 which also contain present tenses)

Response: We have changed to past tense.

Line 124 – Some of the countries excluded from the study would have met the criteria as described (ie. Malawi). Please be more specific if there were questions/data specifically needed for the analysis that excluded some countries.

Response: We have added two more countries(Malawi and Mozambique). 

Line 130 -the sentence on weighted sounds like a conclusion from the previous sentence. Weighting needs to be described briefly as it’s own step in the analysis, not as part of the study population

Response: We described weighting under data analysis. Here we only indicated the weighted sample included in the study as shown in Table 1. 

Line 159 – remove ‘also’.

Response: Done

Line 167 – why would ‘almost every day’ be moderate and not high exposure, when the other questions of ‘at least once a week’ count as high for each type of media? If the ‘3’ refers to the individual question (ie. 3 on a scale of 0-2), then that should be made clear, and then the high media category would become >4 rather than 4-6.

Response: We have revised this accordingly. 

Line 172 – sample weights should be described as ‘women’s sample weights’, but just as ‘Weighting’

Response: We have revised accordingly

Line 180 – ‘pooled the data to create a single dataset’ is more accurate

Response: We have revised the sentence accordingly. 

Results

Line 189 – use present tense ‘present’, not ‘presented’ (sorry, a bit confusing but now that you are presenting your results in the paper, you use the present tense)

Response: Corrected

Lines 215-222 – The description of your analysis belongs in your methods section. The results section should only include results.

Response: We have deleted the sentences belonging to the methods. 

Line 216-217 – is Model 1 not just your unadjusted analysis presented above in Table 3? 

Response: Table 3 is a descriptive table. The p-values are from Pearson chi-square. 

Where is the univariable analysis that shows the association between each of the covariates and the outcome? They should be included in your Table 3, with additional columns – it’ll likely take up an entire page in landscape orientation but that’s fine. 

Response: We have revised table 3. It now shows association between the covariates and the outcomes.

For model 2, did you identify which covariates were significantly associated with transmission in before fitting them in the model? Or did you just include everything in the model? Need to explain how multivariates were selected

Response: We included all these variables based on previous studies indicating they were significantly associated with testing.

Line 220 – “Direction of effect” cannot be established from a cross-sectional study design. You cannot claim cause and effect. You can only demonstrate an association between variables.

Response: We have revised accordingly.

---

## [Editor Report · Decision Letter 1]

14 Oct 2020

PONE-D-20-21800R1

Prenatal care coverage and correlates of HIV testing in sub-Saharan Africa: Insight from demographic and health surveys of 16 countries

PLOS ONE

Dear Dr. Ajayi,

Thank you for submitting your manuscript to PLOS ONE. After careful consideration, we feel that it has merit but does not fully meet PLOS ONE’s publication criteria as it currently stands. Therefore, we invite you to submit a revised version of the manuscript that addresses the points raised during the review process.

Thank you submitting your revised manuscript. The manuscript has benefitted from the change in focus and the additional analyses included as Supplementary Data. However,  some important issues raised by the initial reviewers have only been addressed in part. I have summarized the outstanding issues below.

We look forward to receiving your revised manuscript.

Kind regards,

Emma K. Kalk

Academic Editor

PLOS ONE

Additional Editor Comments (if provided):

PONE-D-20-21800R1

Prenatal care coverage and correlates of HIV testing in sub-Saharan Africa: Insight from demographic and health surveys of 16 countries

The authors have submitted a revised manuscript the focus of which has been broadened to include multiple factors associated with antenatal HIV testing uptake in sub-Saharan Africa. This widening of scope addresses some of the main concerns with the initial report in that 1) the study design (cross-sectional surveys) did not support a causal relationship between HIV-knowledge and test uptake; 2) confounding and collinearity, particularly related to country-specific PMTCT policies which may include mandatory/opt-out ANC education and testing, and country-specific uptake of ANC services in general. However, some of the issues raised have not been completely addressed.

General

The Title, Abstract and Discussion have been revised to reflect the change in focus i.e. “coverage of HIV testing during pregnancy and also examine the factors associated with uptake.” The Results section remains focused on Knowledge. As noted in previous reviewer comments: attendance of ANC=pretest counselling=knowledge. Attendance of ANC=test. Do you have any data on ANC uptake?

Please apply consistency with respect to numbers i.e. numerals or words.

Specific

Line 84-84. ANC testing is standard of care in Botswana and South Africa (every 3 months in the latter) with an opt-out policy. HIV education is part of the testing process.

line 108. Add: “According to the COM-B model….” You have described the summary of the COM-B model suggested by the reviewer. Please could you apply this to the analysis.

Line 134. You appropriately use the weighted datasets from the DHS surveys. As noted before, please provide a brief description of what this means in the text of Methods.

Line 143. Why did you select “knowledge” as the main explanatory variable of interest? As noted, this is likely colinear with local PMTCT policy and ANC attendance itself, both of which may be more relevant. Perhaps “knowledge” could be included as one of several variables. You should also note that COUNTRY was a variable included in your models.

Line 158. The Media Exposure classification is still unclear. You present 2 classifications: low = 0; moderate =1-3 or 1-4; high=4-6 or 5-9. Do the latter apply to Zambia only? If it isn’t possible to apply a single classification, please be explicit as to which system is applied to which country.

Line 169. Detail on weightings as noted above. You only need provide this once.

FIGURES: Has Figure 1 been deleted? There is no longer a legend. Please relabel all the Figures starting at 1 in the text and Figure legends.

Table 2. What is the Total column? It looks like the total number of women included i.e. your denominator? The cause of confusion is use of the comma which you haven’t used in the frequency columns. Please be consistent with the numbers.

Line 220. Typo – “never” is floating.. “never-married”? Proportion is singular so “proportion …. was higher…”

Line 224. Sentence incomplete.

Line 236. “To examine the factors associated with uptake of HIV testing during pregnancy, we fitted two models and presented the results in (Table 4).”

Line 245 – 246. Please could you include COUNTRY as a variable in both models in Table 4. ANC uptake and PMTCT policy, which are captured in the COUNTRY variable, are key factors in ANC testing uptake. I note stratified results are presented in Supplementary tables.

Perhaps mention that you have looked at ANC uptake (Supplementary 7). Is there a reason you couldn’t use this as a variable in the models?

Line 301. Delete “also”

Lines 320-324. The focus of the manuscript has broadened. I feel that emphasis on Knowledge limits the discussion. As noted, Knowledge is colinear with many other variables which may be more important. The study proposition is no longer defining a causal relationship between knowledge and testing (this is not possible with the study design). Lines 330-332 are more important.

Line 332. The COM-B theoretical model (mentioned in the Introduction) could be very useful here as it would address some of the issues with confounding. It would be useful to discuss it’s application to your analysis here, as you do with the HBM model.

Line 335. Does media exposure differ by urban-rural area of SES status?

---

## [Author Response · Author response to Decision Letter 1]

23 Oct 2020

PONE-D-20-21800R1

Prenatal care coverage and correlates of HIV testing in sub-Saharan Africa: Insight from demographic and health surveys of 16 countries

Dear Editor, 

Many thanks for giving us another opportunity to revise our manuscript. We believe your constructive comments have further helped us improve our paper. Please see below our response to all comments you have raised. 

Best Regards

Anthony

The authors have submitted a revised manuscript the focus of which has been broadened to include multiple factors associated with antenatal HIV testing uptake in sub-Saharan Africa. This widening of scope addresses some of the main concerns with the initial report in that 1) the study design (cross-sectional surveys) did not support a causal relationship between HIV-knowledge and test uptake; 2) confounding and collinearity, particularly related to country-specific PMTCT policies which may include mandatory/opt-out ANC education and testing, and country-specific uptake of ANC services in general. However, some of the issues raised have not been completely addressed.

Response: We thank the editor for the comment. We have broadened our focus, and have now emphasised other factors and particularly the role of antenatal care attendance. The opt-out strategy is widely implemented in SSA, including in west African countries; however, other challenges exist which we have highlighted in the manuscript. We did not indicate a causal relationship between knowledge of MTCT and testing in the manuscript. Also, we did not include antenatal care attendance in the Model, given that the rate of testing is 0% among women who did not attend antenatal care. However, the country as a variable is a proxy for the rate of antenatal care utilisation rate as well as the difference in the implementtion of the Opt-out strategy. We also presented table S7 to highlight the role of antenatal care attendance further. Countries with a high rate of antenatal care utilisation generally have a high rate of prenatal care testing. 

General

The Title, Abstract and Discussion have been revised to reflect the change in focus i.e. “coverage of HIV testing during pregnancy and also examine the factors associated with uptake.” 

Response: we thank the editor for the positive feedback

The Results section remains focused on knowledge. 

Response: We have further revised the result section by deleting the results presented on knowledge and Table 2 and figure 3. 

As noted in previous reviewer comments: attendance of ANC=pretest counselling=knowledge. Attendance of ANC=test. Do you have any data on ANC uptake?

Response: The statement that attendance of ANC=pretest counselling=knowledge. Attendance of ANC=test, even though could be implied, is not completely supported by our analysis. The correlation coefficient of knowledge of MTCT and testing was 0.6. It is not too high to be left out of our Model. It is important to note that so many women are still not tested even though they received antenatal care across SSA and especially in West Africa. In fact, ensuring the all women who receive ANC are tested will increase the prevalence of testing significantly, especially in West Africa. 

There is data on ANC; however, the rate of testing among those who did not attend ANC is 0%, not allowing for estimating the odds ratio. However, the point on the role of ANC is well made in the manuscript with the information presented in Table S7. Countries with a high rate of ANC use had a high rate of testing of pregnant women. The country was included in our Model, which we consider to be a proxy for differences in the policy context and ANC use across SSA. 

Please apply consistency with respect to numbers i.e. numerals or words.

Response: we have used numeral throughout

Specific

Line 84-84. ANC testing is standard of care in Botswana and South Africa (every 3 months in the latter) with an opt-out policy. HIV education is part of the testing process.

Response: We have added a sentence to reflect this insight. We thank the editor for this comment. 

line 108. Add: “According to the COM-B model….” You have described the summary of the COM-B model suggested by the reviewer. Please could you apply this to the analysis.

Response: We have explained how this Model informed our analysis under the variable measure section.

Line 134. You appropriately use the weighted datasets from the DHS surveys. As noted before, please provide a brief description of what this means in the text of Methods.

Response: We thank the editor for the feedback. We have indicated what it means. 

Line 143. Why did you select “knowledge” as the main explanatory variable of interest? As noted, this is likely colinear with local PMTCT policy and ANC attendance itself, both of which may be more relevant. Perhaps “knowledge” could be included as one of several variables. You should also note that COUNTRY was a variable included in your models.

Response: we have revised our manuscript such that this could no longer be implied. Also, we have shown the results for country as we included country in our model. Please see Table 3. 

Line 158. The Media Exposure classification is still unclear. You present 2 classifications: low = 0; moderate =1-3 or 1-4; high=4-6 or 5-9. Do the latter apply to Zambia only? If it isn’t possible to apply a single classification, please be explicit as to which system is applied to which country.

Line 169. 

Response. We have revised the description as indicated. 

Detail on weightings as noted above. You only need provide this once.

Response: we have deleted the sentence.

FIGURES: Has Figure 1 been deleted? There is no longer a legend. Please relabel all the Figures starting at 1 in the text and Figure legends.

Response: No, we have added the label

Table 2. What is the Total column? It looks like the total number of women included i.e. your denominator? The cause of confusion is use of the comma which you haven’t used in the frequency columns. Please be consistent with the numbers.

Response: we have deleted table 2 so as not to make the paper focus on knowledge of MTCT. 

Line 220. Typo – “never” is floating.. “never-married”? Proportion is singular so “proportion …. was higher…”

Response- we have corrected this. Thank you

Line 224. Sentence incomplete.

Response: we have completed the sentence

Line 236. “To examine the factors associated with uptake of HIV testing during pregnancy, we fitted two models and presented the results in (Table 4).”

Response: sentence has been revised.

Line 245 – 246. Please could you include COUNTRY as a variable in both models in Table 4. ANC uptake and PMTCT policy, which are captured in the COUNTRY variable, are key factors in ANC testing uptake. I note stratified results are presented in Supplementary tables.

Perhaps mention that you have looked at ANC uptake (Supplementary 7). Is there a reason you couldn’t use this as a variable in the models?

Response: we included country in the Model, we have now shown the result in the paper. ANC was not included because the prevalence of testing among women who did not receive ANC was 0%. We could not estimate odds ratio given that there is nothing to reference. We have indicated the result presented in Table S7 in results. 

Line 301. Delete “also”

Response: deleted 

Lines 320-324. The focus of the manuscript has broadened. I feel that emphasis on knowledge limits the discussion. As noted, knowledge is colinear with many other variables which may be more important. The study proposition is no longer defining a causal relationship between knowledge and testing (this is not possible with the study design). 

Response: we agree with this comment and have revised accordingly. Again, we did not infer a causal relationship; rather, we indicated that knowledge is associated with the uptake of testing. We did test for collinearity, as indicated earlier. 

Lines 330-332 are more important.

Line 332. The COM-B theoretical Model (mentioned in the Introduction) could be very useful here as it would address some of the issues with confounding. It would be useful to discuss it’s application to your analysis here, as you do with the HBM model.

Response: we have discussed the relevance of COM-B

Line 335. Does media exposure differ by urban-rural area of SES status?

Response: we expected that media exposure would differ by place of residence and socio-economic status, given that access to media is generally more in urban areas.

---

## [Editor Report · Decision Letter 2]

26 Oct 2020

Prenatal care coverage and correlates of HIV testing in sub-Saharan Africa: Insight from demographic and health surveys of 16 countries

PONE-D-20-21800R2

Dear Dr. Ajayi,

We’re pleased to inform you that your manuscript has been judged scientifically suitable for publication and will be formally accepted for publication once it meets all outstanding technical requirements.

Kind regards,

Emma K. Kalk

Academic Editor

PLOS ONE
---

## [Editor Report · Acceptance letter]

29 Oct 2020

PONE-D-20-21800R2 

Prenatal care coverage and correlates of HIV testing in sub-Saharan Africa: Insight from demographic and health surveys of 16 countries 

Dear Dr. Ajayi:

I'm pleased to inform you that your manuscript has been deemed suitable for publication in PLOS ONE. Congratulations! Your manuscript is now with our production department. 

Kind regards, 

on behalf of

Dr. Emma K. Kalk 

Academic Editor

PLOS ONE